# Peoples' understanding, acceptance, and perceived challenges of vaccination against COVID-19: A cross-sectional study in Bangladesh

**Alak Paul**[1]*, **Dwaipayan Sikdar**[2], **Janardan Mahanta**[3], **Sanjib Ghosh**[3], **Md. Akib Jabed**[4], **Sujat Paul**[5], **Fahmida Yeasmin**[1], **Suranjana Sikdar**[6], **Bishawjit Chowdhury**[5], **Tapan Kumar Nath**[7]

1 Department of Geography and Environmental Studies, University of Chittagong, Chittagong, Bangladesh, 2 Department of Biochemistry and Molecular Biology, University of Chittagong, Chittagong, Bangladesh, 3 Department of Statistics, University of Chittagong, Chittagong, Bangladesh, 4 Research Associate, Center for Participatory Research and Development (CPRD), Dhaka, Bangladesh, 5 Department of Medicine, Chittagong Medical College, Chittagong, Bangladesh, 6 Department of Microbiology, University of Chittagong, Chittagong, Bangladesh, 7 University of Nottingham Malaysia, Semenyih, Malaysia

* paul.alak@cu.ac.bd

**Data Availability Statement:** All relevant data are within the paper and its Supporting Information files.

## Abstract

In order to eliminate COVID-19, many countries provided vaccinations. However, success depends on peoples' knowledge levels and rates of acceptance. But, previous research on this topic is currently lacking in Bangladesh. This cross-sectional study aimed at to investigate Bangladeshi peoples' knowledge, acceptance, and perception of challenges regarding COVID-19 vaccines. Quantitative data were collected using an online survey (n = 1975) and face-to-face interviews (n = 2200) with a pre-tested structured questionnaire. In addition, seven open-ended interviews were conducted with health experts regarding challenges of vaccination. Binary logistic regression analyses were conducted to assess the association between explanatory and dependent variables. Effect size was estimated to understand the magnitude of relationship between two variables. Of 4175 respondents, 92.6% knew about COVID-19 vaccines, while only 37.4% believed vaccines to be effective in controlling COVID-19. Nearly 46% of respondents believed that COVID-19 vaccines have side-effects, and 16.4% of respondents believed that side-effects could be life-threatening. Only 60.5% of respondents indicated that they would receive the COVID-19 vaccine. Out of 1650 respondents (39.5%) who did not intend to receive the vaccine, 948 (57.4%) believed that they would be naturally protected. Regressions results indicated that men had higher rates of knowledge regarding the vaccine. In addition, rural respondents demonstrated lower knowledge regarding the vaccine. Furthermore, education had a significant association with knowledge of COVID-19 vaccines. Respondents with university education had more knowledge regarding the vaccine (Odds ratio, OR = 29.99; 95% confidence interval, CI 11.40–78.90, effect size 1.88; p = 0.01) and correct dosage (OR 27.34; 95% CI 15.25–49.00, effect size 1.83; p = 0.01). However, women (OR 1.16; 95% CI 0.96–1.40, effect size 0.08) and rural (OR 1.24; 95% CI 1.07–1.44, effect size 0.12; p = 0.01) respondents were more

**Funding:** The author(s) received no specific funding for this work.

**Competing interests:** The authors have declared that no competing interests exist.

enthusiastic regarding receiving the COVID-19 vaccine. Higher educated respondents showed higher probability of receiving the vaccine. Those who believed in the effectiveness of the COVID-19 vaccine were 11.57 times more interested (OR 11.57; 95% CI 8.92–15.01, effect size 1.35; p = 0.01) in receiving the vaccine. Open-ended interviews identified several challenges toward successful COVID-19 vaccination. Mass awareness creation, uninterrupted supply, equitable distribution, and sectoral coordination were suggested to achieve at least 70% immunization across the country.

## Introduction

In over one year, approximately 200 million people worldwide were infected with COVID-19, of which 4.3 million have died [1]. The COVID-19 pandemic had a significant negative impact on people's socio-economic lives [2–6]. Various non-therapeutic measures were adopted to contain COVID-19; however, vaccines were required to control the pandemic and save lives [7]. Vaccination is the most effective method of preventing the spread of an infectious disease [8–10]. The development of a safe and effective vaccine requires time; however, researchers developed several vaccine candidates against COVID-19 in a short time [11]. By December 2020, 57 COVID-19 vaccine candidates were in clinical trials, with some candidates demonstrating efficacy of as high as 95% in preventing symptomatic COVID-19 infections [12]. Many countries started administering the vaccine in 2021. Successful immunization should reduce the global burden of illness and death, as long as the majority of people receive vaccines [6, 13]. It was recommended that 70% of the population be vaccinated in order to achieve herd immunity against COVID-19 [14]. Xiao and Wong [15] identified several factors responsible for wider vaccine acceptance, including the safety and efficacy of the vaccine, adverse health outcomes, misconceptions about the need for vaccination, lack of trust in the health system, and lack of knowledge among the community on vaccine-preventable diseases. In many countries and regions of the world, large variability in the acceptance of COVID-19 vaccines was reported [16, 17]. Peoples' distrust and unwillingness to receive the vaccine could hinder the management and outcomes of COVID-19 inoculation [18].

In addition to peoples' acceptance, several management-related issues could affect smooth immunization against COVID-19. For instance, Pfizer-BioNtech vaccine must be stored at a temperature of -70-degrees Celsius, which is challenging for technologically poor countries [19]. Other challenges may include the availability of staff to oversee the vaccines, equipment, and vaccinators, data systems to track advancement, and methods of informing people regarding the second dose [7, 20]. Moreover, prioritizing a particular group for vaccination and equitable distribution of vaccine could be further obstacles [21].

Bangladesh reported its first positive COVID-19 case on March 8, 2020 and the total number of positive cases rose to 1249,484 as of August 3, 2021, including 20,685 deaths [22]. The Government of Bangladesh signed the Memorandum of Understanding with the Serum Institute of India to receive 30 million doses of the Oxford-AstraZeneca (Covishield) vaccine. In addition, Bangladesh would receive 11 million doses of the COVID-19 vaccine from the Global Alliance for Vaccines and Immunization (GAVI) under the COVID-19 Vaccines Global Access (COVAX), an initiative of the World Health Organization [23]. Peoples' knowledge about COVID-19 vaccines, level of acceptance, and perception of immunization management issues is likely to affect smooth inoculation and achieving herd immunity. However, previous research on this topic is currently lacking. Therefore, the objectives of this study were to

explore the understanding and acceptance of COVID-19 vaccines, and perceptions regarding immunization challenges among people in Bangladesh. This study hypothesized that knowledge of COVID-19 vaccines and acceptance is influenced by respondents' gender, education and residence, such as urban versus rural. The findings of this study provide useful information, which health officials may consider for achieving the expected level of COVID-19 immunization in the country.

## Methods

### Study design and setting

A cross-sectional design was adopted to explore knowledge and acceptance of COVID-19 vaccines, and perception on challenges of vaccination among Bangladeshi people aged 18 years and above. Both quantitative and qualitative data were collected through an online survey, face-to-face interviews, and in-depth interviews. As questionnaires appropriate for this study, particularly the Bangladesh context, were not available, an original questionnaire was developed for quantitative data collection. A questionnaire was drafted based on the authors' prior research experience and mass media information, which was reviewed by two experts and underwent a preliminary evaluation (pre-test) with 70 respondents. Based on experts' comments and pre-test feedback, some questions were eliminated or rephrased for clarity. The final structured questionnaire consisted of 21 multiple choice questions in four categories: (1) socio-demographic information, including four questions, (2) knowledge of COVID-19 vaccines, including nine questions, (3) acceptance of COVID-19 vaccines, including five questions, and (4) COVID-19 vaccine management, including three questions (S1 Table). The questionnaire was prepared in English and translated into *Bengali*, the national language of Bangladesh.

### Questionnaire validation

In order to validate the questionnaire, six relevant experts were asked to assess its relevance for this study. The relevance of a questionnaire has been widely used to measure content validity [24–27]. All experts reported that the items (questions) and responses were relevant to achieve the objectives of this study.

### Study population and inclusion criteria

Study population of this research was Bangladeshi nationals living in the country during the study period. The inclusion criteria were being a Bangladeshi resident aged 18 years or over and living in Bangladesh at the time of the survey.

### Data collection technique

A Google form link was shared through social media, including Facebook, Messenger, and WhatsApp, and respondents were asked to share the link with friends and relatives. In addition, the form was shared through emails with different groups. The form was available from January 24 to February 6, 2021. A total of 1975 responses were collected.

Face-to-face interviews were conducted by research assistants using the same questionnaire and following standard operation procedures, such as wearing masks, safe distancing, and maintaining proper hygiene. Interviews targeted respondents with limited Internet access. Following a convenience sampling strategy [28], respondents were selected in an ad-hoc fashion based on accessibility, following the same inclusion criteria as above. A total of 2200 respondents were interviewed. The response rate was approximately 75%. Each interview took

approximately 10 minutes and was conducted at road sides, small bazars, urban slums, agri-farms, and tea stalls.

Following COREQ guidelines [29], seven in-depth telephone interviews were conducted with officials of health administration of Bangladesh (civil surgeons, divisional health directors, medical college hospital directors, and public health experts) to collect qualitative data. One of the authors, who had social medicine expertise, conducted the interviews. Interviews were guided by a check-list consisting of open-ended questions related to the challenges, acceptance, confidence in vaccine management, and the government's future plans regarding vaccine safety. Interviewees were selected deliberately based on their engagement in the public health sector.

### Ethical considerations

This study was approved by the ethical review committee of Chittagong Medical College, Bangladesh (Memo No. CMC/PG/2021/130). Participation was voluntary and anonymous, and respondents were informed that they could withdraw from the survey at any time. Informed consent was obtained electronically through the form prior to beginning the questionnaire. For face-to-face interviews, verbal consent was obtained after the aims of the study were explained, and for in-depth interviews, informed verbal consent to participate in this study was obtained before each interview.

### Sample size and power calculation

A total of 4175 responses (1975 online and 2200 face-to-face) were obtained, which was satisfactory at the 95% confidence level with a ±5% margin of error [30]. Israel [31] suggested a sample size of 400 at a 95% confidence level with a ±5% precision level. For $\alpha = 0.05$ and a hypothesized proportion of 0.5, the power of a sample size of 4175 is 1.0 (calculated using the SigmaXL statistical tool). It was assumed that a sample size of 4175 for this study is adequate to generalize the study findings.

### Outcome measures

This study examined two major outcome measures. First one was knowledge of COVID-19 vaccination of respondents (five dependent variables namely heard about the COVID-19 vaccine, believe that vaccine control COVID-19, dose, side effects, and type of side effects) using gender, age, resident and education as explanatory variables among the respondents. Second outcome measure was opinion of acceptance of COVID-19 vaccine of respondents (five dependent variables namely like to take COVID-19 vaccine; reason-protected from COVID-19; reason-take and control transmission; Bangladesh produces the COVID-19 vaccine, would you take it; and possible side effects and temporary protection) using gender, age, resident, education, believe that vaccination can control COVID-19, dose and type of side effects as explanatory variables.

### Data analysis and reporting

Descriptive statistics (frequency and percentage) of responses were estimated. Binary logistic regression analyses were conducted to assess the association between explanatory and dependent variables. There were few assumptions of running logistic regression. First, logistic regression does not require a linear relationship between the dependent and explanatory variables. Second, the error terms (residuals) do not need to be normally distributed and homoscedasticity is not required. Finally, the dependent variable in logistic regression is not measured on an

interval or ratio scale. Odds ratio (OR) with a 95% confidence interval (CI) was used to assess the strength of association, and p-values of less than 0.05 were considered statistically significant. Following Chinn [32], effect size (expressed as "Cohen's d") was estimated to understand the magnitude of relationship between two variables. Explanatory variables with more than two categories were grouped into two "yes" and "no" categories (S2 Table). To check model fitness, the omnibus chi-square test was used and all models were found to be highly significant (S3 Table). The significance value of less than 0.05 indicated that the current model outperformed the null model. In all models, significant chi-square values indicated that the fitted model was better than the null model. In order to report and describe qualitative data, COREQ guidelines were followed. Statistical analyses were conducted using the Statistical Package for the Social Sciences (SPSS) version 16 (SPSS for Windows, Version 16.0. Chicago, SPSS Inc.).

## Results

### Socio-demographic characteristics of respondents

Due to the sampling methods, gender balance could not be ensured. Of 4175 respondents, there were more men (n = 2723; 65.2%) than women (Table 1). The distribution of respondents in five age classes was consistent to the age structure of Bangladesh [33]. There was a predominant number of university students (n = 1185; 28.4%), likely due to internet and social media access. As expected, most respondents (n = 2404; 57.6%) were from urban areas, possibly due to higher education and internet access.

### Respondents' knowledge on COVID-19 vaccines and level of acceptance

The results indicated that approximately 93% of the respondents heard or knew about COVID-19 vaccines, largely from television news (68.7%) and social media (38.7%; Table 2). The government of Bangladesh took extensive measures to ensure awareness through

**Table 1. Socio-demographic information of respondents.**

| Variables | Frequency (n = 4175) | Percentage |
|---|---|---|
| Gender | | |
| Men | 2723 | 65.2 |
| Women | 1452 | 34.8 |
| Age (years) | | |
| 18–30 | 1547 | 37.1 |
| 31–40 | 1196 | 28.6 |
| 41–50 | 862 | 20.6 |
| 51–60 | 387 | 9.3 |
| Above 60 | 183 | 4.4 |
| Education | | |
| Illiterate | 770 | 18.4 |
| Primary | 810 | 19.4 |
| Secondary | 643 | 15.4 |
| Higher Secondary | 767 | 18.4 |
| University | 1185 | 28.4 |
| Residence | | |
| City/Urban | 2404 | 57.6 |
| Rural | 1771 | 42.4 |

**Table 2. Respondents' knowledge on COVID-19 vaccination.**

| Variables | Frequency of responses | Percentage of responses |
|---|---|---|
| Do you know/hear about COVID-19 vaccine? (N = 4175) | | |
| Yes | 3864 | 92.6 |
| No | 311 | 7.4 |
| How do you know about COVID-19 vaccine? (Multiple answers are allowed) (n = 3864)* | | |
| Newspapers | 940 | 24.3 |
| Television news | 2654 | 68.7 |
| Social media | 1494 | 38.7 |
| Friends or Colleagues | 931 | 24.1 |
| Family members | 523 | 14.8 |
| Do you know that COVID-19 vaccines are available in some countries? (n = 3864) * | | |
| Yes | 2859 | 74.0 |
| No | 1005 | 26.0 |
| Do you know/believe that vaccination can control COVID-19? (N = 3864) * | | |
| Yes | 1447 | 37.4 |
| No | 420 | 10.9 |
| Not sure | 1997 | 51.7 |
| Do you know how many doses require for proper vaccination? (n = 3864) * | | |
| One Dose | 189 | 4.9 |
| Two Doses | 806 | 20.9 |
| Not sure | 2869 | 74.2 |
| Do you think that COVID-19 vaccines would have some side effects? (n = 4175) | | |
| Yes | 1930 | 46.2 |
| No | 225 | 5.4 |
| Not sure | 2020 | 48.4 |
| Which type of side effects may arise in the body after vaccination? (n = 4175) | | |
| Primary side effects (fever, headache, vomiting, etc.) | 1037 | 24.8 |
| Serious side effects (life threatening) | 683 | 16.4 |
| No idea | 2455 | 58.8 |
| Which age group should be prioritized in receiving corona vaccine? (n = 4175) | | |
| Old people | 1700 | 40.7 |
| Adult People | 460 | 11.0 |
| Children | 260 | 6.2 |
| Adolescent | 24 | 0.6 |
| All | 1278 | 30.6 |
| Not sure | 453 | 10.9 |
| Which professional group should be prioritized in receiving vaccine? (n = 4175) (maximum three answers are allowed) | | |
| Health care workers | 2588 | 62.0 |
| People suffering prolong disease | 1496 | 35.8 |
| Non COVID-19 but hospitalized patients | 493 | 11.8 |
| Politicians | 231 | 5.5 |
| Bureaucrats | 179 | 4.3 |
| Security Personals | 1013 | 24.3 |
| Teachers | 183 | 4.4 |
| Students | 328 | 7.9 |
| Not sure | 693 | 16.6 |

* Those answered 'No' in response to question no 1 were skipped for this question.

television channels. In addition, respondents (74%) knew that COVID-19 vaccines were available in many countries. However, respondents had little knowledge on the effectiveness of vaccines, with only 37.4% indicating that vaccines prevent COVID-19. Only 20.9% of respondents correctly mentioned that two doses of the vaccine are required for proper immunization. Furthermore, 53.8% of respondents were not aware of any side-effects of COVID-19 vaccines, and 58.8% of respondents did not know about the types of side-effects. About 41% of respondents reported that older people should receive the COVID-19 vaccine, and 62% believed that healthcare workers should be prioritized.

The results indicated that only 60.5% of respondents were interested in receiving a COVID-19 vaccine (Table 3), with the majority of them (74.6%) stating that COVID-19 vaccines would protect them from infection and 36.8% opined that the vaccine would reduce or control virus transmission. Respondents who indicated that they did not wish to receive a COVID-19 vaccine reported several reasons, including belief that the vaccine is "not necessary, I am fine and naturally protected" (57.4% respondents), "possible side effects and temporary protection" (39.9%), and "religious reasons" (18.5%). Those interested in receiving a COVID-19 vaccine preferred the Pfizer-BioNTech vaccine (24.4%), while 50.5% did not know about vaccine options. However, 88.8% of the respondents wanted to receive a Bangladeshi COVID-19 vaccine, if available.

**Table 3. Respondents' opinion on acceptance of COVID-19 vaccine.**

| Variables | Frequency of Responses | Percentage of responses |
|---|---|---|
| Do you like to take COVID-19 vaccine? (n = 4175) | | |
| Yes | 2525 | 60.5 |
| No | 1650 | 39.5 |
| Why do you like to take COVID-19 vaccine? (Multiple answers are allowed) (n = 2525) | | |
| I will be protected from COVID-19 | 1884 | 74.6 |
| Government has suggested to take and control transmission | 277 | 11.0 |
| Helps to reduce COVID-19 fears | 282 | 11.2 |
| Helps to reduce/control COVID-19 transmission | 930 | 36.8 |
| Which vaccine would you prefer to take? (n = 2525) | | |
| Pfizer/BioNTech, USA | 617 | 24.4 |
| Moderna, USA | 98 | 3.9 |
| AstraZeneca, UK | 141 | 5.6 |
| Sinovac, China | 38 | 1.5 |
| Sinopharm, China | 2 | 0.1 |
| Sputnik, Russia | 53 | 2.1 |
| Covishield/Serum institute, India | 238 | 9.4 |
| Covaxin/Bharat Biotech, India | 62 | 2.5 |
| No Idea | 1276 | 50.5 |
| If Bangladesh produces the COVID-19 vaccine, would you take it? (n = 2525) | | |
| Yes | 2241 | 88.8 |
| No | 284 | 11.2 |
| Those who opined not to take COVID-19 vaccine, what are the reasons? (multiple answer allowed) (n = 1650) | | |
| Religious issue | 305 | 18.5 |
| Possible side effects and temporary protection | 659 | 39.9 |
| Temporary protection | 100 | 6.1 |
| Expensive | 106 | 6.4 |
| Not necessary, I am fine and protected naturally | 948 | 57.4 |

## Factors associated with knowledge of COVID-19 vaccines and level of acceptance: Multivariable logistic regression analysis

Women were less knowledgeable regarding COVID-19 vaccines than men, with women demonstrating 52% lower odds of knowing about COVID-19 vaccines (OR 0.48, 95% CI 0.35–0.67, effect size -0.41; p = 0.01), and 39% lower odds (OR 0.61, 95% CI 0.51–0.73, effect size -0.27; p = 0.01) of having heard about COVID-19 vaccines and their effectiveness (Table 4). In addition, women had 24% lower odds (OR 0.76, 95% CI 0.61–0.95, effect size -0.15; p = 0.05) of knowing about doses of vaccines required for proper immunization, 34% lower odds (OR 0.66, CI 0.56–0.79, effect size -0.23; p = 0.01) of knowing about probable side-effects of vaccines, and 18% lower odds (OR 0.82, CI 0.69–0.97, effect size -0.11; p = 0.05) of knowing about types of side-effects (Table 4). Compared to younger respondents (18–30 years old), older respondents (above 30 years old) heard less about COVID-19 vaccines, with the oldest respondents having 70% lower odds of having heard about COVID-19 vaccines (OR 0.30, 95% CI 0.17–0.52, effect size -0.67; p = 0.01). However, the middle-aged group of the respondents had better knowledge of effectiveness of vaccines against COVID-19 (OR 1.35, 95% CI 1.21–2.07, effect size 0.17; p = 0.01 for the 31-40-year-old group and OR 1.28, 95% CI 1.03–1.60, effect size 0.14; p = 0.05 for the 41-50-year-old group). Respondents between 18–30 years had less knowledge regarding correct doses of COVID-19 vaccines. As expected, respondents in rural areas heard less about vaccines, although the difference was not significant (OR 0.82, 95% CI

**Table 4. Logistic regression on respondents' knowledge of COVID-19 vaccination.**

| Variables | | Odds Ratio (95% CI), effect size[§] | | | | |
|---|---|---|---|---|---|---|
| | | Heard about the COVID-19 Vaccine | Believe that vaccine control COVID-19 | Dose | Side Effects | Type of Side Effect |
| Gender | Men | | | | | |
| | Women | 0.48** (0.35–0.67), -0.41 | 0.61** (0.51–0.73), -0.27 | 0.76* (0.61–0.95), -0.15 | 0.66** (0.56–0.79), -0.23 | 0.82* (0.69–0.97), -0.11 |
| Age | 18–30 | | | | | |
| | 31–40 | 0.83 (0.54–1.27), -0.10 | 1.35** (1.11–1.65), 0.17 | 1.58** (1.21–2.07), 0.25 | 1.11 (0.91–1.35), 0.06 | 1.21 (1.00–1.47), 0.11 |
| | 41–50 | 0.85 (0.55–1.32), -0.09 | 1.28* (1.03–1.60), 0.14 | 2.02** (1.49–2.74), 0.39 | 0.86 (0.69–1.07), -0.08 | 1.30* (1.05–1.61), 0.14 |
| | 51–60 | 0.70 (0.43–1.15), -0.20 | 1.05 (0.79–1.39), 0.03 | 2.20** (1.48–3.27), 0.44 | 0.89 (0.67–1.17), -0.06 | 0.93 (0.70–1.23), -0.04 |
| | 60 above | 0.30** (0.17–0.52), -0.67 | 0.78 (0.51–1.18), -0.14 | 1.90* (1.06–3.40), 0.35 | 0.84 (0.58–1.21), -0.10 | 0.86 (0.59–1.26), -0.08 |
| Resident | City/Urban | | | | | |
| | Rural | 0.82 (0.64–1.06), -0.11 | 1.13 (0.98–1.30), 0.07 | 0.46** (0.38–0.56), -0.43 | 0.72** (0.62–0.82), -0.18 | 0.88 (0.77–1.01), -0.07 |
| Education | Illiterate | | | | | |
| | Primary | 1.76** (1.31–2.36), 0.31 | 1.01 (0.80–1.28), 0.01 | 1.76 (0.91–3.42), 0.31 | 1.37** (1.09–1.72), 0.17 | 1.70** (1.34–2.15), 0.29 |
| | Secondary | 3.97** (2.58–6.11), 0.76 | 1.20 (0.93–1.55), 0.10 | 7.21** (3.97–13.07), 1.09 | 2.11** (1.65–2.70), 0.41 | 2.70** (2.10–3.47), 0.55 |
| | Higher Secondary | 6.41** (3.48–11.78), 1.03 | 1.37* (1.05–1.78), 0.17 | 13.01** (7.21–23.48), 1.42 | 3.00** (2.32–3.89), 0.61 | 3.29** (2.53–4.29), 0.66 |
| | University | 29.99** (11.40–78.90), 1.88 | 1.43** (1.10–1.86), 0.20 | 27.34** (15.25–49.00), 1.83 | 4.76** (3.68–6.15), 0.86 | 4.35** (3.35–5.64), 0.81 |

Note

§: Small effect if Cohen's $|d| \leq 0.2$; moderate effect if Cohen's $d$ $0.2 < |d| \leq 0.5$; large effect if Cohen's $|d| > 0.5$

0.64–1.06, effect size -0.11). Surprisingly, rural respondents showed more knowledge of the effectiveness of vaccines. Level of education was significant. Respondents with university-level education heard more about COVID-19 vaccines (OR 29.99, 95% CI 11.40–78.90, effect size 1.88; p = 0.01) and had more knowledge of effectiveness (OR 1.43, 95% CI 1.10–1.78, effect size 0.20; p = 0.01), correct doses (OR 27.34, 95% CI 15.25–49.00, effect size 1.83; p = 0.01), possible side-effects (OR 4.76, 95% CI 3.68–3.89, effect size 0.86; p = 0.01), and types of side-effects (OR 4.35, 95% CI 3.35–5.64, effect size 0.81; p = 0.01).

Although there was no significant association, women were more likely to receive the COVID-19 vaccine (OR 1.16, 95% CI 0.96–1.40, effect size 0.08) than men (Table 5), despite

**Table 5. Logistic regression on respondents' opinion of acceptance of COVID-19 vaccine.**

| Variables | | Odds Ratio (95% CI), effect size§ | | | | |
|---|---|---|---|---|---|---|
| | | Like to take COVID-19 Vaccine | Reason (Protected from COVID-19) | Reason (Take and control transmission) | Bangladesh produces the COVID-19 vaccine, would you take it | Possible side effects and temporary protection |
| Gender | Men | | | | | |
| | Women | 1.16 (0.96–1.40), 0.08 | 0.76** (0.65–0.89), -0.15 | 1.29** (1.08–1.54), 0.14 | 0.89 (0.65–1.21), -0.06 | 1.04 (0.85–1.27), 0.02 |
| Age | 18–30 | | | | | |
| | 31–40 | 1.21 (0.97–1.50), 0.11 | 1.25* (1.04–1.50), 0.12 | 1.59** (1.27–1.99), 0.26 | 1.45 (0.99–2.11), 0.21 | 0.74* (0.59–0.94), -0.17 |
| | 41–50 | 1.44** (1.13–1.83), 0.20 | 1.23* (1.01–1.51), 0.11 | 1.71** (1.34–2.18), 0.30 | 1.07 (0.72–1.60), 0.04 | 0.64** (0.49–0.84), -0.25 |
| | 51–60 | 1.17 (0.87–1.59), 0.09 | 1.03 (0.80–1.33), 0.02 | 1.32 (0.96–1.81), 0.15 | 1.26 (0.73–2.17), 0.13 | 0.72 (0.51–1.02), -0.18 |
| | 60 above | 1.05 (0.69–1.59), 0.03 | 0.71 (0.50–1.01), -0.19 | 0.97 (0.62–1.50), -0.02 | 1.22 (0.52–2.86), 0.11 | 0.57* (0.34–0.95), -0.31 |
| Resident | City/Urban | | | | | |
| | Rural | 1.24** (1.07–1.44), 0.12 | 1.19** (1.05–1.35), 0.10 | 0.81** (0.69–0.93), -0.12 | 2.07** (1.56–2.75), 0.40 | 0.85 (0.72–1.01), -0.09 |
| Education | Illiterate | | | | | |
| | Primary | 0.71** (0.56–0.92), -0.19 | 1.10 (0.89–1.35), 0.05 | 0.64** (0.49–0.83), -0.25 | 0.58* (0.35–0.97), -0.30 | 1.37 (0.99–1.89), 0.17 |
| | Secondary | 1.11 (0.84–1.46), 0.06 | 1.42** (1.13–1.79), 0.19 | 0.99 (0.75–1.31), -0.01 | 0.67 (0.38–1.18), -0.22 | 1.66** (1.18–2.33), 0.28 |
| | Higher Secondary | 1.35* (1.00–1.82), 0.17 | 1.42** (1.11–1.81), 0.19 | 1.47** (1.11–1.94), 0.21 | 0.59 (0.34–1.05), -0.29 | 2.06** (1.46–2.91), 0.40 |
| | University | 1.36* (1.01–1.84), 0.17 | 1.45** (1.15–1.85), 0.21 | 2.02** (1.53–2.65), 0.39 | 0.29** (0.17–0.50), -0.68 | 2.35** (1.67–3.30), 0.47 |
| Believe that vaccination can control COVID-19 | No | | | | | |
| | Yes | 11.57** (8.92–15.01), 1.35 | | | | |
| | Not Sure | 2.55** (2.01–3.24), 0.52 | | | | |
| Dose | Not Sure | | | | | |
| | Two | 1.66** (1.34–2.05), 0.28 | | | | |
| Type of Side Effect | No Idea | | | | | |
| | Some side effect | 0.47** (0.40–0.55), -0.42 | | | | |

Note

§: Small effect if Cohen's |d|≤0.2; moderate effect if Cohen's d 0.2<|d|≤0.5; large effect if Cohen's |d|>0.5

their lower knowledge of vaccines. In addition, women indicated that immunization was more important to control virus transmission (OR 1.29, 95% CI 1.08–1.54, effect size 0.14; p = 0.01) than for personal protection (OR 0.76, 95% CI 0.65–0.89, effect size -0.15; p = 0.01). Respondents in the 41–50 age group were more likely to receive the vaccine (OR 1.44, 95% CI 1.13–1.83, effect size 0.20; p = 0.01) for personal protection (OR 1.23, 95% CI 1.01–1.51, effect size 0.11; p = 0.01) than to control of virus transmission (OR 1.71, 95% CI 1.34–2.18, effect size 0.30; p = 0.01) and were less concerned about side-effects (OR 0.64, 95% CI 0.49–0.84, effect size -0.25; p = 0.01) than the 18–30 age group. Rural respondents had 124% higher odds of receiving the COVID-19 vaccine (OR 1.24, 95% CI 1.07–1.44, effect size 0.12; p = 0.01) for personal protection from COVID-19 (OR 1.19, 95% CI 1.05–1.35, effect size 0.10; p = 0.01). Rural respondents were also more likely to accept the Bangladeshi vaccine (OR 2.07, 95% CI 1.56–2.75, effect size 0.40; p = 0.01), if available. As expected, educated respondents were more interested in receiving the vaccine. Respondents with university-level education had 136% higher odds (OR 1.36, 95% CI 1.01–1.84, effect size 0.17; p = 0.01) of receiving the vaccine for personal protection (OR 1.45, 95% CI 1.15–1.85, effect size 0.21; p = 0.01) and reduction of virus transmission (OR 2.02, 95% CI 1.53–2.65, effect size 0.39; p = 0.01). However, university-educated respondents were more concerned about possible side-effects and temporary protection (OR 2.35, 95% CI 1.46–2.91, effect size 0.47; p = 0.01). Respondents who believed that immunization prevented COVID-19 were 11.57 times more likely to receive the vaccine (OR 11.57, 95% CI 8.92–15.01, effect size 1.35; p = 0.01). Similarly, respondents who knew the correct number of doses of the vaccine had 166% higher odds of receiving the vaccine (OR 1.66, 95% CI 1.34–2.05, effect size 0.28; p = 0.01).

## Respondents' perceptions of challenges of COVID-19 vaccination

The majority of the respondents (53%) believed that security personnel (e.g. Army) would execute immunization properly, while 24.6% believed it would be administered by government hospitals (Table 6).

**Table 6. Respondents' opinion on management of COVID-19 vaccination.**

| Variables | Frequency of responses | Percentage of responses |
|---|---|---|
| In your opinion, how vaccination should be implemented? (n = 4175) | | |
| Through Government Hospitals | 1028 | 24.6 |
| Through Private Clinics | 45 | 1.1 |
| Through Security Forces (e.g. Army, Navy) | 2212 | 53.0 |
| Through NGOs | 330 | 7.9 |
| No idea | 560 | 13.4 |
| What would be the main challenges for COVID-19 vaccination? (multiple answers are allowed) (n = 4175) | | |
| Motivating people to receive vaccine | 1532 | 36.7 |
| Storage and Transport at low temperature | 690 | 16.5 |
| Cost | 1476 | 35.4 |
| Participant selection | 353 | 8.5 |
| Ensuring vaccination safety and Equipment | 819 | 19.6 |
| Coordination between Ministries and field level | 1108 | 26.5 |
| No idea | 1060 | 25.4 |
| Do you think Bangladeshi authority would manage proper distribution of vaccine? (n = 4175) | | |
| No | 1137 | 27.2 |
| Yes | 1476 | 35.4 |
| Not sure | 1562 | 37.4 |

During the interviews, a divisional director of health commented that:

*The health ministry will distribute the vaccine to the public across the country through government hospitals and clinics. We have no plan to include the security personnel in its distribution. Bangladesh has a positive reputation for vaccine management in the world. We hope the Ministry of Health will successfully manage the vaccination program, such as the Expanded Program on Immunization (EPI). We have sufficient vaccine points throughout the country; however, if needed, the government will increase registration and vaccination points.*

Only 35.4% of respondents perceived that Bangladesh authority would manage the distribution of vaccines properly. However, many of the respondents were concerned with challenges of COVID-19 immunization. For instance, 36.7% of respondents reported that motivating the public to receive vaccine would be a major challenge, followed by cost (35.4%), coordination between ministries and field level centers (26.5%), and storage and transport of vaccines at a low temperature (16.5%). A health ministry official stated that:

*Until now, educated and high- and middle-income people and various professionals have shown greater interest in receiving vaccines. But Bangladesh faces some challenges: first, ensuring uninterrupted vaccine supply from the source point, as the government planned, for 130 million people to be vaccinated gradually; second: keeping online registration systems active for this mass immunization program in the whole country; third, in order to encourage the public to get vaccinated, the government needs to create various awareness-building plans and programs; fourth, the country needs a good example of co-ordination between the state mechanisms and other agencies for a successful ending.*

## Discussion

Vaccination against COVID-19 is considered the most effective method to control the COVID-19 pandemic. Successful vaccination and herd immunity among the public depends on peoples' knowledge of COVID-19 vaccines, which may influence vaccine acceptance and proper management. This study attempted to understand Bangladeshi peoples' perceptions of these factors.

### Knowledge and acceptance of COVID-19 vaccines

The results revealed that over 90% of respondents heard about COVID-19 vaccines, although only a small fraction of them knew about the effectiveness, side-effects, and correct doses of the vaccines. This indicated that respondents lacked correct information regarding COVID-19 vaccines, which suggests the need for awareness-building and correct information dissemination programs. An official of the Ministry of Health mentioned:

*We know many people do not have the correct information regarding COVID-19 vaccines and are confused about whether or not to receive the vaccine. The government has already started a campaign through mass media to eliminate false impressions and hesitancy toward vaccination among the public.*

Approximately 46.2% of respondents reported believing that COVID-19 vaccines have several side-effects. Previous studies reported COVID-19 vaccine side-effects, such as pain or discomfort, allergic reactions, swelling, fever, chills, tiredness, and headache within 1–3 days after

the vaccine is administered [34, 35]. Among vaccine recipients globally, few people were reported to experience severe allergic reactions after receiving the COVID-19 vaccine [36]. A higher official of the Ministry of Health stated:

> *We do not have many records or complains about side-effects after vaccination. A few people may face some problems, such as pain, fever, or allergic reactions after receiving the vaccine. We request vaccine recipients to stay at our observation centers for at least 30 minutes after getting the vaccine. We have every preparation to deal with probable side-effects.*

Men demonstrated better knowledge of COVID-19 vaccines than women. This is in line with previous studies [18, 37–40]. Age and level of education had a significant influence on knowledge of COVID-19 vaccines, suggesting that the government should provide extensive motivational programs through various channels, such as mass media and community engagement, to encourage these groups of people to receive the COVID-19 vaccine. Results indicated that only 60.5% of respondents were likely to receive the COVID-19 vaccine. The real rate of vaccination could be lower due to misinformation regarding COVID-19 vaccines and their side-effects on social media, religious beliefs, beliefs in temporary protection, and confidence in self-immunity. Spread of misinformation through multiple channels could considerably affect the acceptance of the COVID-19 vaccine [41]. Therefore, mass awareness programs are crucial in order to create confidence among the public and achieve a 70% vaccination rate to reach herd immunity. Eliminating misconceptions through transparent communication with proper knowledge among people is necessary [38, 42] to reduce the skepticism about vaccination [43]. Higher acceptance rates were reported in India (86.3%), China (89.5%), Malaysia (94.3%), and Indonesia (93.3%) [37, 44, 45]. Countries such as Kuwait (23.6%), Jordan (28.4%), Italy (53.7), Russia (54.9%), Poland (56.3%), the US (56.9%), and France (58.9%) reported lower rates of COVID-19 vaccine acceptance [45]. COVID-19 vaccines acceptance rates below 60% pose a severe problem for controlling the pandemic [45].

Women and rural residents were more interested in receiving the COVID-19 vaccine despite limited knowledge of vaccines. Therefore, the government should provide extensive programs, such as easy vaccination registration and vaccination centers in rural areas, targeting women and rural residents. On the other hand, respondents who believed that vaccine could prevent COVID-19 were approximately 12 times more likely to get vaccinated, confirming that correct information on vaccines is of utmost importance to higher acceptance.

## Challenges of COVID-19 vaccination

Along with socio-demographic characteristics, COVID-19 vaccination may have several challenges, including procurement, distribution, and implementation. The government of Bangladesh drafted a national deployment and vaccination plan to vaccinate 80% of the population in four stages, including ensuring procurement and proper coordination and launching awareness campaigns to address vaccine hesitancy [23]. Bangladesh procured COVID-19 vaccines from several sources and approximately 3.6% population received at least one dose of vaccine and 2.6% population were fully vaccinated by July 4, 2021 [46]. Distribution of vaccines through appropriate organization, setting up priority groups, motivating people, and appropriate infrastructure are important for vaccine management [45]. The results of this study indicated that motivating people to receive the vaccine and coordination among agencies were some of the major challenges to smooth vaccination. Islam and Hossain [47] suggested creating a monitoring team to oversee proper vaccine transportation and storage at the right temperature. An uninterrupted supply chain of vaccines across the country needs to be ensured,

with storage methods following the EPI framework. This can be assumed that the quality of the vaccine degrades if it is not properly preserved, transported, distributed, and administered.

## Study limitation

This study had some limitations. First, respondents were mainly educated people, likely due to access to information and past experience as research participants. People with lower levels of education may feel less confident acting as research participants. Second, respondents participated in this study voluntarily and only those who were interested took part in the survey. As such, gender balance could not be achieved. Third, the participation of rural and urban respondents was unequal, possibly due to urban residents having better access to the Internet. However, these limitations were believed to have no effect on the findings of this study.

## Conclusion

COVID-19 is a deadly disease that requires therapeutic and non-therapeutic solutions. World leaders face challenges in containing COVID-19 through non-therapeutic solutions, with mass vaccination remaining as the primary solution. Knowledge, beliefs, availability, and distribution of the vaccine pose challenges to mass vaccination. This study found mixed responses regarding level of knowledge and acceptance of the COVID-19 vaccine. Raising public awareness and demonstrating positive aspects of vaccination to the public appears to be most effective in increasing the vaccine acceptance rate. Governments, public health officials, and advocacy groups should address hesitancy and build vaccine literacy to encourage the public to accept immunization. COVID-19 immunization program should be implemented across the country to give rural and urban populations equal opportunity to receive the vaccine.

## Supporting information

**S1 Table. Questionnaire on peoples' understanding and acceptability of COVID-19 vaccines, and perceived challenges on successful vaccination in Bangladesh.**
(DOCX)

**S2 Table. Category of logistic regression.**
(DOCX)

**S3 Table. Omnibus tests of model coefficients.**
(DOCX)

## Acknowledgments

We would like to thank the anonymous respondents for volunteering to participate in this study. We also wish to thank the anonymous reviewers and academic editor for their constructive feedback regarding this manuscript. We acknowledge support from the Research and Publication Cell, University of Chittagong in this research.

## Author Contributions

**Conceptualization:** Alak Paul, Dwaipayan Sikdar, Tapan Kumar Nath.

**Data curation:** Janardan Mahanta, Sanjib Ghosh.

**Formal analysis:** Janardan Mahanta, Sanjib Ghosh.

**Investigation:** Md. Akib Jabed, Sujat Paul, Fahmida Yeasmin, Suranjana Sikdar, Bishawjit Chowdhury.

**Methodology:** Alak Paul, Dwaipayan Sikdar, Janardan Mahanta, Tapan Kumar Nath.

**Project administration:** Alak Paul.

**Supervision:** Alak Paul, Dwaipayan Sikdar.

**Writing – original draft:** Alak Paul, Dwaipayan Sikdar.

**Writing – review & editing:** Md. Akib Jabed, Fahmida Yeasmin, Suranjana Sikdar, Tapan Kumar Nath.

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
