## [Decision Letter · Decision Letter 0]

4 Jun 2021

PONE-D-21-12921

Peoples’ understanding, acceptance and perceived challenges towards vaccination against COVID-19: a cross-sectional study in Bangladesh

PLOS ONE

Dear Dr. Paul

Thank you for submitting your manuscript to PLOS ONE. After careful consideration, we feel that it has merit but does not fully meet PLOS ONE’s publication criteria as it currently stands. Therefore, we invite you to submit a revised version of the manuscript that addresses the points raised during the review process.

We look forward to receiving your revised manuscript.

Kind regards,

Nurshad Ali

Academic Editor

PLOS ONE

Journal Requirements:

2.In the Methods section please provide additional information regarding:

1) the participant eligibility criteria

2) A justification for the sample size used in your study, including any relevant power calculations (if applicable).

3) Steps taken to validate the questionnaire.

Furthermore when reporting the results of qualitative research, we suggest consulting the COREQ guidelines: http://intqhc.oxfordjournals.org/content/19/6/349. In this case, please consider including more information on the number of interviewers, their training and characteristics; and please provide the interview guide used. Please provide additional information regarding the type of informed consent taken for the qualitative study (ie written, verbal). Finally please provide additional information regarding how the qualitative data were analysed.

3.We suggest you thoroughly copyedit your manuscript for language usage, spelling, and grammar. If you do not know anyone who can help you do this, you may wish to consider employing a professional scientific editing service.  

Reviewers' comments:

Reviewer's Responses to Questions

**Comments to the Author**

1. Is the manuscript technically sound, and do the data support the conclusions?

Reviewer #1: Yes

Reviewer #2: Yes

2. Has the statistical analysis been performed appropriately and rigorously? 

Reviewer #1: Yes

Reviewer #2: Yes

3. Have the authors made all data underlying the findings in their manuscript fully available?

Reviewer #1: Yes

Reviewer #2: Yes

4. Is the manuscript presented in an intelligible fashion and written in standard English?

Reviewer #1: No

Reviewer #2: Yes

5. Review Comments to the Author

Reviewer #1: Abstract

Back ground

The research gap was not indicated

Results

The proportions of outcome should be written along with 95% CI.

Why did sample size was wide? CI 11.40- 55 78.90; p=0.01) and their correct doses (OR 27.34; 95% CI 15.25-49.00; p=0.01)???

Female (OR 1.16; 95% CI 0.96-1.40) was not statistically significant?

Main body

Materials and Methods

sample size determination was not clear

How do ensure data quality?

pretest was not made

How did you assess outcomes?

The authors didnt check internal consistency of tools

confounding variables was not controlled for possible confounders and identified.

Multicollinearity was not checked

which test is used to check model fitness?

What is the mean( SD) of study subjects?

Results

The proportions of outcome should be written along with 95% CI.

Reviewer #2: this very good article, well written this my commentts

abstract is long and need to be shorten

methods: first please add paragraph about sample size calculation and techniques, regrading study tool please describe steps of construction and validation of questionnaire

regarding analysis, please mention the country of origin and manufactures of SPSS used

please remark correct answer or favorable responses for acceptable and perception questions under table

6. PLOS authors have the option to publish the peer review history of their article (what does this mean?). If published, this will include your full peer review and any attached files.

Reviewer #1: No

Reviewer #2: **Yes: **wafaa Yousif Abdel Wahed

---

## [Author Response · Author response to Decision Letter 0]

8 Jul 2021

We (the authors) have addressed all editorial and reviewers’ comments carefully in the revised manuscript. We show our best gratitude to both reviewers and the academic editor for inviting us to revise this paper. We believe that PLOS ONE will find this article a timely and quality enough for publication. Thanks.

---

## [Decision Letter · Decision Letter 1]

28 Jul 2021

PONE-D-21-12921R1

Peoples’ understanding, acceptance, and perceived challenges of vaccination against COVID-19: A cross-sectional study in Bangladesh

PLOS ONE

Dear Dr. Paul,

Thank you for submitting your manuscript to PLOS ONE. After careful consideration, we feel that it has merit but does not fully meet PLOS ONE’s publication criteria as it currently stands. Therefore, we invite you to submit a revised version of the manuscript that addresses the points raised during the review process.

We look forward to receiving your revised manuscript.

Kind regards,

Nurshad Ali

Academic Editor

PLOS ONE

Journal Requirements:

Reviewers' comments:

Reviewer's Responses to Questions

**Comments to the Author**

1. If the authors have adequately addressed your comments raised in a previous round of review and you feel that this manuscript is now acceptable for publication, you may indicate that here to bypass the “Comments to the Author” section, enter your conflict of interest statement in the “Confidential to Editor” section, and submit your "Accept" recommendation.

Reviewer #1: (No Response)

Reviewer #2: (No Response)

2. Is the manuscript technically sound, and do the data support the conclusions?

Reviewer #1: No

Reviewer #2: Yes

3. Has the statistical analysis been performed appropriately and rigorously? 

Reviewer #1: Yes

Reviewer #2: Yes

4. Have the authors made all data underlying the findings in their manuscript fully available?

Reviewer #1: Yes

Reviewer #2: Yes

5. Is the manuscript presented in an intelligible fashion and written in standard English?

Reviewer #1: No

Reviewer #2: Yes

6. Review Comments to the Author

Reviewer #1: Abstract

Background: State rephrase the objective of study

Methods: indicate outcome measure

Result : express effect size along with 95% CI for magnitude of outcomes

Avoid redundancy eg OR, ODDS RATIO??

Main body

Study design, setting, source population, study pop, inclusion criteria, sample size determination, sampling method, data collection technique, data quality control and data analysis should be stated clearly. identify the type of logistic regression analysis, What are the assumptions of logistic regression model, Have you checked the interactions? indicate the value?

Results

Write your current finding in logical order.

Express effect size along with 95% CI for magnitude of outcomes.

Reviewer #2: in the method section please add at first a small paragraph described the study design and setting,

reliability assessment of questionnaire, please provide if do any measure to assess that, provide the cronbach-alfa of the used scale

sample size : you write that uggested a sample size of 400 at a 95% confidence level with a ±5% precision

210 level [31]. For α = 0.05 and a hypothesized proportion of 0.5, ) but you take a very large size . you can you other assumption to suit your large size,

please provide us paragraph that adjust for this size and mention the sampling technique used

7. PLOS authors have the option to publish the peer review history of their article (what does this mean?). If published, this will include your full peer review and any attached files.

Reviewer #1: No

Reviewer #2: **Yes: **Wafaa Y. Abdel wahed

---

## [Author Response · Author response to Decision Letter 1]

5 Aug 2021

A file has been added in the online system named 'Response to Reviewers'.

---

## [Editor Report · Decision Letter 2]

9 Aug 2021

Peoples’ understanding, acceptance, and perceived challenges of vaccination against COVID-19: A cross-sectional study in Bangladesh

PONE-D-21-12921R2

Dear Dr. Paul,

We’re pleased to inform you that your manuscript has been judged scientifically suitable for publication and will be formally accepted for publication once it meets all outstanding technical requirements.

Kind regards,

Nurshad Ali

Academic Editor

PLOS ONE

---

## [Editor Report · Acceptance letter]

13 Aug 2021

PONE-D-21-12921R2 

Peoples’ understanding, acceptance, and perceived challenges of vaccination against COVID-19: A cross-sectional study in Bangladesh 

Dear Dr. Paul:

I'm pleased to inform you that your manuscript has been deemed suitable for publication in PLOS ONE. Congratulations! Your manuscript is now with our production department. 

Kind regards, 

on behalf of

Dr. Nurshad Ali 

Academic Editor

PLOS ONE